# Olfactory marker protein (OMP) regulates formation and refinement of the olfactory glomerular map

Dinu F. Albeanu[1], Allison C. Provost[2,3], Prateek Agarwal[2,3], Edward R. Soucy[2,3], Joseph D. Zak [2,3] & Venkatesh N. Murthy [2,3]

Inputs from olfactory sensory neuron (OSN) axons expressing the same type of odorant receptor (OR) converge in the glomerulus of the main olfactory bulb. A key marker of mature OSNs is olfactory marker protein (OMP), whose deletion has been associated with deficits in OSN signal transduction and odor discrimination. Here, we investigate glomerular odor responses and anatomical architecture in mice in which one or both alleles of *OMP* are replaced by the fluorescent synaptic activity reporter, synaptopHluorin. Functionally heterogeneous glomeruli, that is, ones with microdomains with distinct odor responses, are rare in $OMP^{+/-}$ mice, but occur frequently in $OMP^{-/-}$ mice. Genetic targeting of single ORs reveals that these microdomains arise from co-innervation of individual glomeruli by OSNs expressing different ORs. This glomerular mistargeting is locally restricted to a few glomerular diameters. Our studies document functional heterogeneity in sensory input within individual glomeruli and uncover its anatomical correlate, revealing an unexpected role for OMP in the formation and refinement of the glomerular map.

[1] Cold Spring Harbor Laboratory, Cold Spring Harbor, Cold Spring Harbor, NY 11724, USA. [2] Department of Molecular and Cellular Biology, Harvard University, Cambridge, MA 02138, USA. [3] Center for Brain Science, Harvard University, Cambridge, MA 02138, USA. These authors contributed equally: Dinu F. Albeanu, Allison C. Provost, Prateek Agarwal. Correspondence and requests for materials should be addressed to V.N.M. (email: vnmurthy@fas.harvard.edu)

Olfactory sensory neurons (OSNs) reside in the olfactory epithelium where they bind odorants[1] and send projections to the olfactory bulb (OB) where axons expressing the same odorant receptor (OR) converge to form a single glomerulus on the OB surface[2–4]. This one receptor–one glomerulus rule implies that the odor sensitivities of OSNs throughout a given glomerulus are homogeneous, defined by the odor binding properties of the OR. This overarching rule of the OB provides the foundation for a stereotyped map across animals[5–7].

Glomeruli, which tile the OB, are arranged in a stereotyped pattern[2,6–8], and the two hemifields of glomeruli on the OB surface form mirror images of each other[6,8–11] such that each OR-defined glomerulus will appear twice on the surface of each hemisphere of the OB. OSNs synapse onto principal (mitral and tufted) neurons and interneurons in the OB[11,12]. Principal neurons of the OB receive excitatory input from a single glomerulus and convey output signals that are informed by the odor sensitivities of this parent glomerulus, but are also influenced by indirect inhibitory inputs from other glomeruli, top-down feedback and neuromodulatory signals[11,12].

Intra-glomerular heterogeneity may arise from mixed OSN input to individual glomeruli. Heterogeneous glomeruli are normally rare[7,13], but molecular or activity perturbations to olfactory sensory neurons can increase the frequency of anatomically mixed glomeruli[14–16]. Each OB hemisphere in the mouse is thought to have around 3500 glomeruli[17], which would be more than sufficient to accommodate two glomeruli per OR type, one per hemifield, given ~1100 OR types. Several identified OR-specific OSNs appear to converge onto more than one glomerulus per hemifield[7,18–21], which skews the ratio of OR number to glomerular number and could lead to heterogeneous OR glomeruli in wild-type mice.

All mature OSNs in vertebrates express a protein called olfactory marker protein (OMP), encoded by a small intronless gene[22–25]. OMP is implicated in olfactory signal transduction[26], with a role in determining the latency and duration of odor responses by OSNs[27]. OMP has also been proposed to aid the maturation of OSNs, for example, speeding up the restriction of OR gene expression to a single type[26]. A functional role for OMP was also suggested by a recent study in which glomeruli of OMP-null mice respond to a wider range of odorants than expected[28]. In addition, OSNs of OMP-null mice do not always terminate in the glomerular layer, but sometimes project deeper into the external plexiform layer[29]. The exact molecular function of OMP remains elusive, but pharmacological studies indicate that it functions upstream of cyclic adenosine monophosphate (cAMP) production in OSN cilia, possibly as a phosphodiesterase inhibitor[30]. Combined with OMP presence in the axon terminals of OSNs[31,32] and the regulatory action of cAMP on OSN axon guidance[33,34], this observation raises the possibility that OMP may regulate axon path finding and glomerular map formation.

We set out to probe the functional and anatomical glomerular heterogeneity in OMP-synaptopHluorin (spH) mice[35], in which spH is knocked into the OMP locus in the genome, replacing the coding sequence of OMP. This renders OMP-spH homozygous mice ($OMP^{-/-}$) knockouts for OMP, whereas OMP-spH heterozygotes ($OMP^{+/-}$ mice) express one copy of the gene. Using functional imaging, as well as anatomical reconstructions, we find that OMP is necessary to preserve the one receptor–one glomerulus rule in the adult olfactory bulb.

## Results

### Absence of OMP alters odor responses

It has recently been shown that glomerular responses in the OMP-null mice exhibit wider tuning curves than in OMP heterozygotes[36], but these experiments utilized a very limited array of odors. This led us to ask, with a diverse odor panel, how glomerular odor representations are changed in $OMP^{-/-}$ mice. We explored this question in OMP-spH mice that have SpH, a synaptic activity reporter, knocked into the OMP locus and expressed in all OSNs. Using widefield imaging of spH signals on the dorsal surface of the bulb to a panel of 99 odorants (Supplementary Table 1), we found that single glomeruli, as determined by their spherical macrostructure (Fig. 1a), in the $OMP^{-/-}$ mouse responded to more odors when compared to $OMP^{+/-}$ mice ($8.5 \pm 0.5$ in $OMP^{+/-}$ and $9.7 \pm 0.6$ in $OMP^{-/-}$; mean $\pm$ standard error of the mean; $p < 0.005$, Kolmogorov–Smirnov (K-S) test; Fig. 1c, d, left). At the same time, the total number of active glomeruli in the imaged region of the dorsal surface was higher in $OMP^{-/-}$ mice ($72.6 \pm 3.3$) than in $OMP^{+/--}$ mice ($53.9 \pm 1.3$); $p < 0.005$, K-S test; Fig. 1b, right. Also, the number of glomeruli responding to single odors in the panel nearly doubled in $OMP^{-/-}$ compared to $OMP^{+/-}$ mice ($8.2 \pm 0.5$ in $OMP^{+/-}$ and $16.2 \pm 0.6$ in $OMP^{-/-}$; $p < 0.01$, K-S test; Fig. 1c, d, center). These differences in response patterns are indicative of pronounced changes in glomerular activity patterns in the $OMP^{-/-}$ genetic background.

In order to better understand the functional changes in glomerular responses, we also examined whether the loss of OMP leads to changes in the macro-organization of glomeruli. Does the increased number of responding glomeruli indicate the presence of more glomeruli in $OMP^{-/-}$ mice? Using fixed tissue slices, we counted the number of glomeruli present in the most central OB slices. Glomerular borders were clearly indicated by 4′,6-diamidino-2-phenylindole (DAPI) staining of the vast number of juxtaglomerular cell nuclei surrounding the glomerulus[37]. We found that the total number of glomeruli did not significantly change between $OMP^{+/-}$ and $OMP^{-/-}$ mice ($99 \pm 3.4$ and $104 \pm 1.8$ glomeruli per OB slice, respectively; Fig. 1a, left; $p > 0.65$, K-S test). Moreover, the size of glomeruli in the $OMP^{+/-}$ and $OMP^{-/-}$ mice was very similar ($4303 \pm 149\ \mu m^2$ and $4266 \pm 149\ \mu m^2$, respectively Fig. 1c, d, right; $p > 0.3$, K-S test), corresponding to glomerular average diameters of 74.0 μm and 73.7 μm, consistent with previous studies[38,39]. Together, our results indicate that the macrostructure of the glomerulus is relatively unperturbed in the absence of OMP. The increased promiscuity of glomerular odor responses, but lack of change in the anatomical macro-organization of glomeruli, led us to hypothesize that OMP plays a role in shaping the organization of OSN axons within individual glomeruli.

### Functional microdomains in $OMP^{-/-}$ mice

We asked how responses within single glomeruli change in the OMP-null background. We used two-photon laser scanning microscopy (2PLSM) to examine the microstructure of glomerular responses to odors in OMP-spH mice[35,40]. Glomerular activity in each field of view was probed serially with a panel of 32 odorants (Supplementary Table 1). A spatially contiguous region of resting fluorescence surrounded by dark voids was taken to be an individual glomerulus (Fig. 2a).

Odors evoked robust increases in fluorescence intensity as described before via widefield microscopy[6,28,35,36], as well as using 2PLSM[13,35,41]. When we examined odor responses in $OMP^{-/-}$ mice (Fig. 2a), we frequently observed fluorescence increases within sub-regions of individual glomeruli (Fig. 2b, c). At higher magnification, we noticed that different odors activated distinct parts of a single glomerulus (Fig. 2c, Supplementary Movies 1,2). Analysis of the time course of responses in different sub-regions confirmed that different odors differentially activate at least three regions within the example glomerulus shown in Fig. 2b.

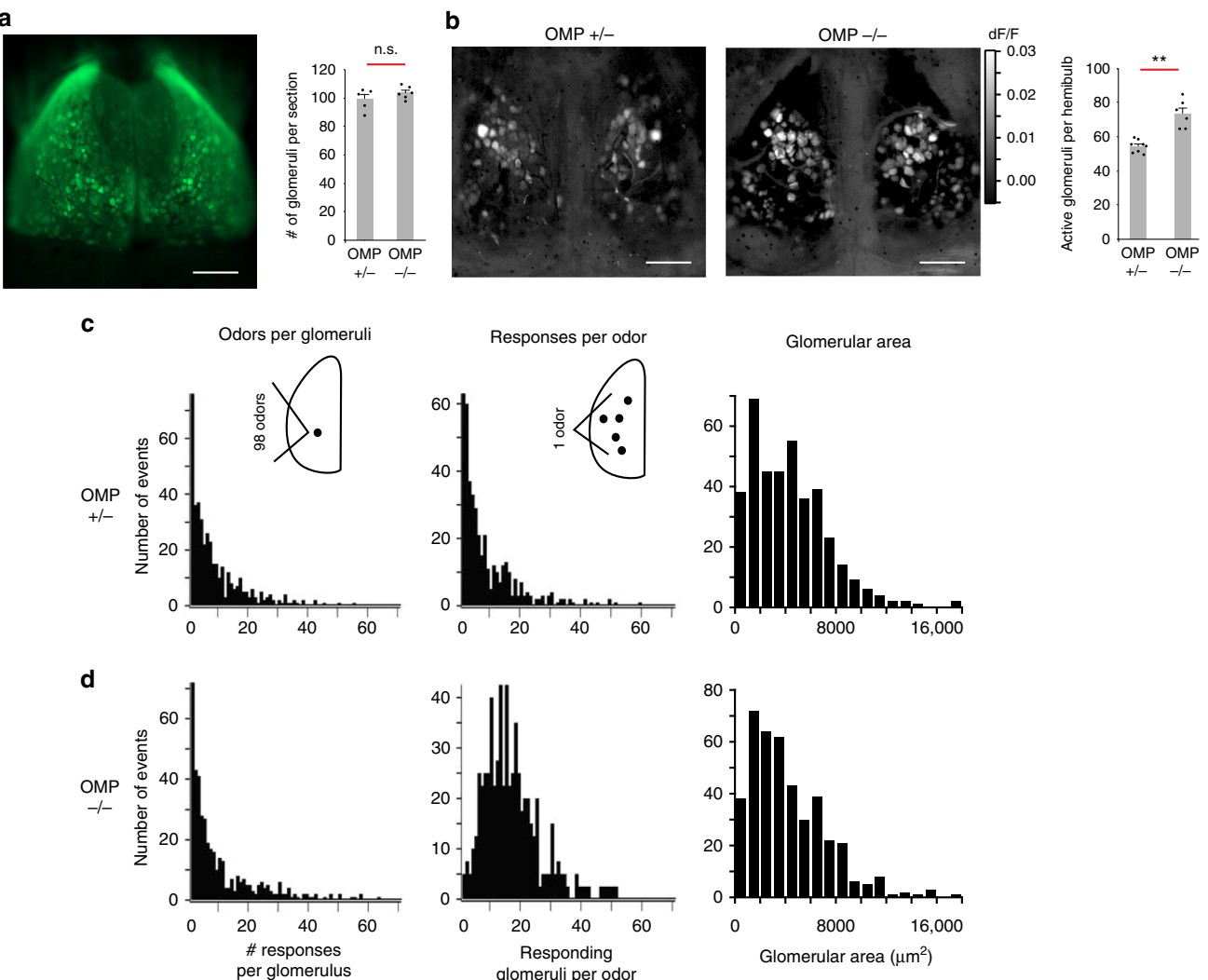

**Fig. 1** Absence of OMP alters odor responses, but preserves macro-organization of the OB. **a** (Left) Widefield view of dorsal surface of an $OMP^{+/-}$ mouse shows labeled glomeruli from the resting fluorescence of spH. (Right) Quantification of the number of glomeruli per coronal section in central sections of the OB. Shown are 99 ± 3.44 glomeruli per coronal section for $OMP^{+/-}$ mice (mean ± SEM, n = 5 slices from 2 mice) and 104 ± 1.8 glomeruli per coronal section for $OMP^{-/-}$ (mean ± SEM, $n$ = 6 slices from 2 mice). **b** (Left) Maximum projection glomerular odor response maps for exemplar $OMP^{+/-}$ and $OMP^{-/-}$ mice to a panel of 99 odorants. (Right) Average number of responsive glomeruli per hemibulb to the odor panel (Supplementary Table 1). Shown are 53.9 ± 1.3 glomeruli per hemibulb for $OMP^{+/-}$ mice (mean ± SEM, $n$ = 8 hemibulbs) and 72.6 ± 3.3 glomeruli per hemibulb for $OMP^{-/-}$ (mean ± SEM, $n$ = 6 hemibulbs). **c**, **d** (Left) Histograms of the number of odors to which a single glomerulus responds. (Center) Histograms of the number of glomeruli activated by an individual odor in the panel. (Right) Histograms of glomerular area in central coronal sections of the OB in $OMP^{+/-}$ and $OMP^{-/-}$ mice. n.s. indicates difference is not statistically significant, **$p < 0.005$, Kolmogorov–Smirnov test; scale bar, 500 µm

To obtain an objective view of the response heterogeneity within the imaged glomeruli (Fig. 2a), we used principal component analysis (PCA) to classify all pixels in the imaged area according to their similarity (Methods). If a glomerulus is entirely homogeneous in its responses to odors, all the pixels are expected to be similar to each other. On the other hand, if different groups of pixels evoke distinct response profiles to odors, they will be separated out by PCA, even if they are not spatially contiguous. Here, each pixel was assigned an integrated odor response value (average change in fluoresence over resting fluoresence (dF/F)) for each of the 32 odors, so each pixel was associated with an odor response vector.

Pixels above the signal threshold were projected on the plane determined by two of the first three principal components, and it became apparent that they defined three functional clusters (Fig. 2d). Interestingly, the three functional clusters mapped onto three contiguous spatial microdomains within the glomerulus and

matched the spatial domains activated by individual odorants (Fig. 2e). This correspondence was revealed by correlating the average odor response vectors of the three PCA-identified functional clusters (Fig. 2d, bottom) to the odor responses of each pixel in the field of view above the noise threshold (see Methods). Superimposing the three PCA-identified pixel clusters (using an RGB (red, green, and blue) color code) fully reconstructed the glomerular anatomy, as assessed from the resting fluorescence image (Fig. 2f), leaving no empty regions.

To understand the three-dimensional structure of this heterogeneous glomerulus, we performed PCA again on single pixel responses for a different optical plane (20 µm more superficial) within the same glomerulus (Fig. 2f). Three functional domains were also apparent in this second optical plane. These domains were very similar in their odor tuning to the ones identified in the first plane (pixel correlation > 0.9), but they occupied different relative spatial proportions. In the superficial optical plane, the

'blue' microdomain dominated the glomerulus response, pushing aside 'green' response territory. Similar exemplar data are shown for several other glomeruli in Fig. 2g and Supplementary Figure 1 (also see Supplementary Movies 3, 4). The spatial structure within the functional microdomains identified suggest that these clusters are due to different types of OSN inputs innervating the same glomerulus. Another argument in favor of multi-OSN-type

glomerular innervation is that in many instances, such as the z-stack shown in Supplementary Figure 2 (also Supplementary Movies 5–7), we could follow different bundles of OSN fibers of slightly different resting fluorescence levels that converge within the glomerulus (sampling every 2 µm along the z-axis), and further relate them functionally to glomerular microdomains by eliciting differential responses with the panel of odorants.

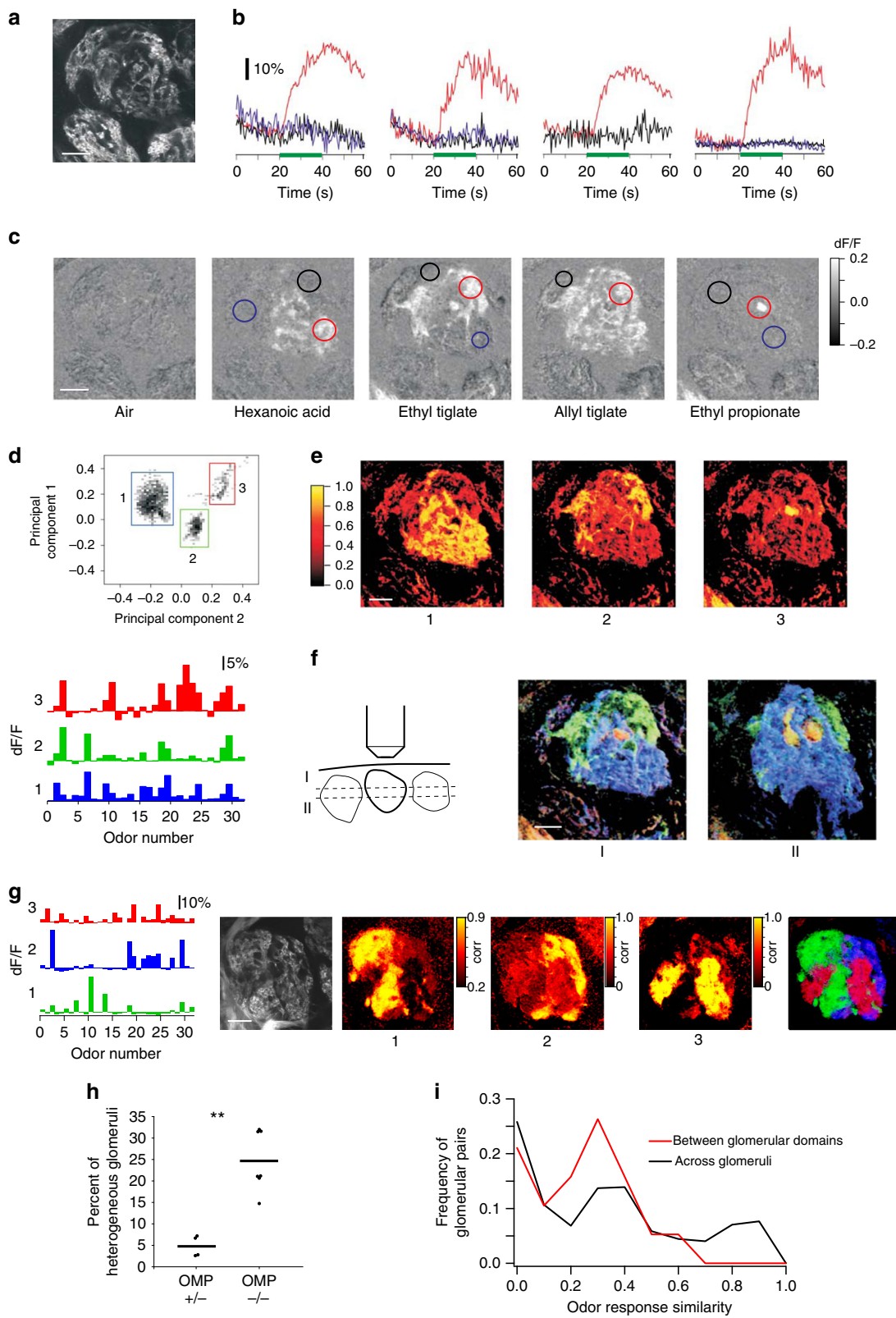

To determine the frequency of occurrence of mixed glomeruli, we outlined anatomical boundaries of glomeruli using the resting fluorescence images. We then counted the total number of active glomeruli in the imaged region. Functionally heterogeneous glomeruli were identified in both $OMP^{+/-}$ and $OMP^{-/-}$ animals by visual inspection followed by PCA (Methods). Functionally mixed glomeruli occurred at significantly higher rate (24.7 ± 6.9%, $n = 7$ bulbs, 195 active glomeruli) in the $OMP^{-/-}$ mice, compared to $OMP^{+/-}$ mice (4.8 ± 2.5%, $n = 4$ bulbs, 118 active glomeruli, Fig. 2h, Mann–Whitney $U$ test, $p = 0.0058$). We further calculated the pairwise odor response similarity between functional domains observed within the anatomical glomeruli versus across random pairs of glomeruli (including those that appear heterogeneous) in the fields of view sampled in $OMP^{-/-}$ mice. The intra-glomerular microdomains were no more similar to each other in their odor tuning than random pairs of distinct anatomical glomeruli (Fig. 2i).

To further investigate how the apparent mixing of multiple types of OR fibers alters the glomerular map in $OMP^{-/-}$ mice, we sampled larger fields of view tiling the surface of the bulb. We found that functionally fragmented glomeruli were generally located near glomeruli that responded uniformly to the odorant panel and were functionally matched to one microdomain of the mixed glomeruli (Fig. 3a). In many instances, within a few glomerular lengths, we also identified additional glomerular fragments that had similar tuning to the odors in our panel (Fig. 3b and Supplementary Fig. 3). By contrast, in $OMP^{+/-}$ mice, the occurrence of functionally heterogeneous glomeruli and of glomerular duplicates was significantly lower (Fig. 2h, Supplementary Fig. 4).

The specific gene-targeted $OMP$-$spH$ mouse line used in our study did not allow us to monitor glomerular spH responses in $OMP^{+/+}$ mice, since $spH$ is knocked-in to the $OMP$ locus replacing the $OMP$ gene (therefore $OMP^{+/+}$ mice will not express spH). We were, however, able to image glomeruli in $OMP^{+/+}$ mice expressing a different sensor of activity (GCaMP3) in the OSNs[42]. In response to a smaller odor panel (15 odorants), we found that the percent of heterogeneous glomeruli was 4.8 ± 2.3% (5 bulbs, 97 glomeruli, Supplementary Fig. 5). Since the $OMP$-GCaMP3 experiments were part of a different study[55], the exact identity and number of odors used were different from those used here; yet, this is in remarkable agreement with the number of heterogeneous glomeruli in heterozygous OMP-spH mouse data (4.8 ± 2.5%).

**Anatomical microdomains are prominent in $OMP^{-/-}$ mice.** We hypothesized that the functional microdomains observed in

$OMP^{-/-}$ mice were due to anatomical heterogeneity. Specifically, we predicted that multiple types of OSNs, as defined by the ORs they express, converge onto a single glomerular structure. To test this hypothesis, we used mouse lines with single labeled glomeruli, $M72$-red fluorescent protein (RFP)[43] and $P2$-LacZ[20], crossed into the $OMP$-$spH$ background, and compared the glomerular convergence in $OMP^{+/-}$ mice to $OMP^{-/-}$ littermates. In heterogeneous glomeruli, we expect to observe genetically labeled OR-specific OSN axons intermixed with unlabeled OSN axons (if they were expressing a different, untargeted OR).

As expected, glomeruli from $OMP^{+/-}$ mice were generally homogeneous (Fig. 4a). By contrast, in $OMP^{-/-}$ mice, partially filled, heterogeneous glomeruli were frequently identified (Fig. 4b). In order to quantify heterogeneity within a single glomerulus, we determined the overlap between P2 β-galactose signal (pseudocolored red in Fig. 4a) and OMP-spH signal (green in Figs 2, 3, see Methods). Glomerular borders were determined by the cluster of nuclei or juxtaglomerular cells, stained by DAPI (Fig. 4a). We were then able to express glomerular homogeneity as the ratio of number of spH-positive pixels that were also P2 positive and the number of total spH-positive pixels. This quantifies the percentage of pixels in a glomerulus that express our target gene. P2-positive fibers frequently ramified into anatomical microdomains of the target glomerulus of $OMP^{-/-}$ mice, on average filling 61% of the glomerulus ($n = 51$ glomeruli) compared with 82% in $OMP^{+/-}$ mice ($n = 40$ glomeruli) ($p < 0.001$, K-S test; Fig. 4c).

In the $M72$-$RFP$ line, we also observed that $M72$-$RFP$ $OMP^{+/-}$ mice often exhibited filled, homogeneous glomeruli (Fig. 4d). Interestingly, the $M72$-$RFP$ line showed a distinct phenotype of heterogeneity, with discrete subdomains sometimes visible (Fig. 4e, top), but often displaying wider, more diffuse borders (Fig. 4e, bottom row). We did not amplify RFP signals using secondary conjugates, which led to poorer detection efficiency than for spH signals. Therefore, the percent overlap even in $OMP^{+/-}$ mice was low, and we relied on relative measures. M72-RFP glomeruli had significantly higher homogeneity fraction in $OMP^{+/-}$ than in $OMP^{-/-}$ mice (54%, $n = 19$ glomeruli versus 41%, $n = 18$ glomeruli, $p < 0.05$, K-S test; Fig. 4f).

**Duplicate glomeruli are more frequent in $OMP^{-/-}$ mice.** In addition to microdomains within single glomeruli, we also found functional evidence of duplicate glomeruli, found in close proximity to the target sites (Fig. 3). This duplication could arise from OSNs expressing a particular OR terminating in multiple glomeruli in the vicinity of each other. Gene-targeted glomeruli are found in well-defined locations on the OB

**Fig. 2** Analysis of functional glomerular heterogeneity in $OMP^{-/-}$ glomeruli. **a** Raw two-photon fluorescence of a glomerulus in an $OMP^{-/-}$ mouse. **b**, **c** Example time courses and average responses (dF/F) of the glomerulus in (**a**) to 4 different stimuli. Colored circles mark regions of interest and correspond to color of traces in (**b**). Green horizontal line marks stimulus delivery. **d** (Top) Projection of all pixels in (**a**) on two principal components (PC1 vs. PC2) reference coordinates. PCA was performed on the odor response spectra of each pixel. (Bottom) Average odor response spectra of three functional clustered identified in glomeruli from (**a**). Color scale units are correlation coefficients. **e** Spatial correlograms showing the spatial distribution of the three pixel clusters identified via PCA in (**d**) within the example glomerulus (**a**) were obtained by correlating the average response vectors of the identified clusters to the odor responses of individual pixels in the field of view. Numbers mark correlation maps corresponding to each functional cluster in (**d**). Note that each cluster corresponds to a spatially contiguous area we refer to as a microdomain. **f** (Left) Cartoon depicting two optical imaging planes (z) within the example glomerulus (**a**) at two different depths (I and II, 20 μm apart). (Center and Right) RGB color scheme overlays of the correlograms determined as in (**e**) for the two optical planes sampled within the glomerulus. Blue corresponds to image 1 in (**e**), green to image 2, red to image 3. **g** Three spatial microdomains in a second example glomerulus in the $OMP^{-/-}$ mouse. (Left) Average odor response spectra of functional clusters identified via PCA. (Right) Resting fluorescence, single pixel correlograms, and their overlay in an RGB scheme. Color scale units are correlation coefficient. **h** Summary graph of the average frequency of glomerular heterogeneity recorded in $OMP^{+/-}$ and $OMP^{-/-}$ mice (4.8 ± 2.5%, $n = 4$ bulbs versus 24.7 ± 6.9%, $n = 7$ bulbs, $p = 0.0058$, Mann–Whitney $U$ test; horizontal bars represent the mean, dots individual animals). **i** Histogram of pairwise odor response similarity between functional domains within the same anatomical glomerulus (red) and between different glomeruli (black) in $OMP^{-/-}$ mice. Results are presented as means ± SEM; **$p < 0.001$; scale bar, 20 μm

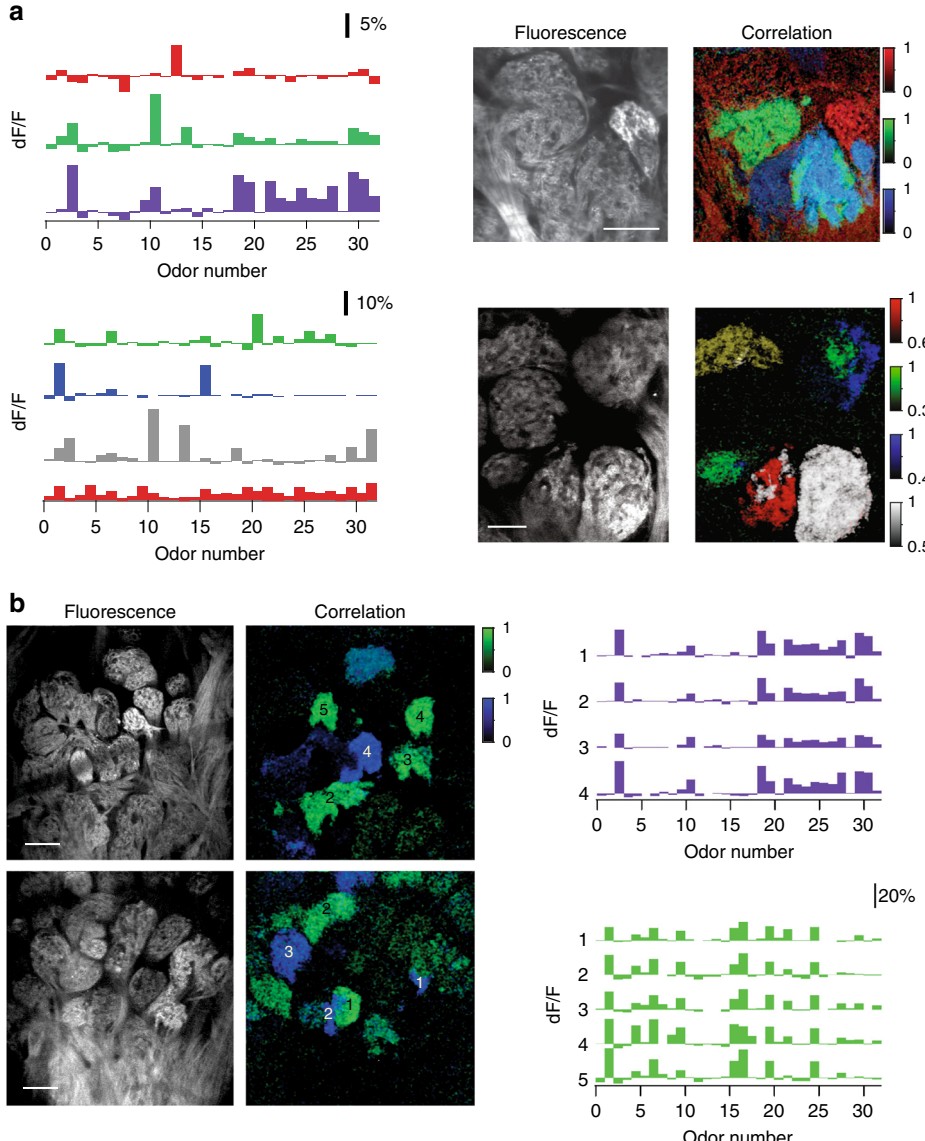

**Fig. 3** Local functional glomerular duplications in *OMP⁻/⁻* mice. **a** (Left) Example average odor response spectra of PCA-identified functional clusters corresponding to glomerular microdomains from two different fields of view (top vs. bottom). (Right) Corresponding resting fluorescence and overlay of correlograms. Note the presence of functionally homogeneous and heterogeneous glomeruli adjacent to each other. For example, in the lower panel, six anatomically identifiable glomeruli are discernable. The correlograms (Right) indicate that green microdomains are shared between two close-by anatomical glomeruli. White and red microdomains mix within one anatomical glomerulus. The white cluster is also present as a spatial-functional homogeneous adjacent glomerulus. Color scale units are correlation coefficient. Scale bar, 50 µm. **b** Two partially overlapping example fields of view in one *OMP⁻/⁻* mouse. (Left) Resting glomerular fluorescence. (Center) Correlograms: colors correspond to functionally matched glomeruli or subglomerular microdomains. (Right) Odor response spectra for the green and blue glomeruli. Numbers correspond to the location of the functionally similar glomeruli in the sampled fields of view. Color scale units are correlation coefficient. Scale bar, 80 µm

surface, and there are usually two, one per hemifield, though low-frequency local duplications have been reported even in adult wild-type mice[7]. To determine the number of labeled glomeruli in each hemifield, we imaged OBs in whole-mount preparation (Fig. 5a). On both the *OMP⁻/⁻* and *OMP⁺/⁻* background, M72-RFP glomeruli were always located on the dorsal surface, segregating to both the medial and lateral hemifields (Fig. 5a, b, left panels). We never observed ectopic glomeruli that terminated well beyond the expected target region. As expected, in the OMP⁺/⁻ background M72-RFP glomeruli were usually limited to one per hemifield (Fig. 5a, top), but the M72-RFP glomerulus appeared frequently duplicated in *OMP⁻/⁻* mice (Fig. 5a, bottom). Overall, glomerular duplication in the *M72-RFP* line occurred more frequently in

*OMP⁻/⁻* mice than *OMP⁺/⁻* littermates, with an average of 1.36 ± 0.07 (mean ± SEM, n = 76 hemibulbs) and 1.05 ± 0.03 (n = 56 hemibulbs) M72-RFP glomeruli per hemifield, respectively (p < 0.001, Wilcoxon rank-sum test; Fig. 5b). Importantly, duplicate glomeruli were always located within a few average glomerular diameters of each other. In a subset of experiments, where the entire dorsal and medial surfaces of the OB were visualized, we were able to estimate the distance between pairs of duplicate glomeruli (Avg. 152.9 ± 23.2 µm SEM in *OMP⁻/⁻* mice, n = 12 glomerular pairs).

The P2 glomerulus has been shown to form duplicates on a wild-type background[20], and these glomeruli are positioned proximally to one another, and can be difficult to discern in whole-mount brains. Because of this, we quantified the number of

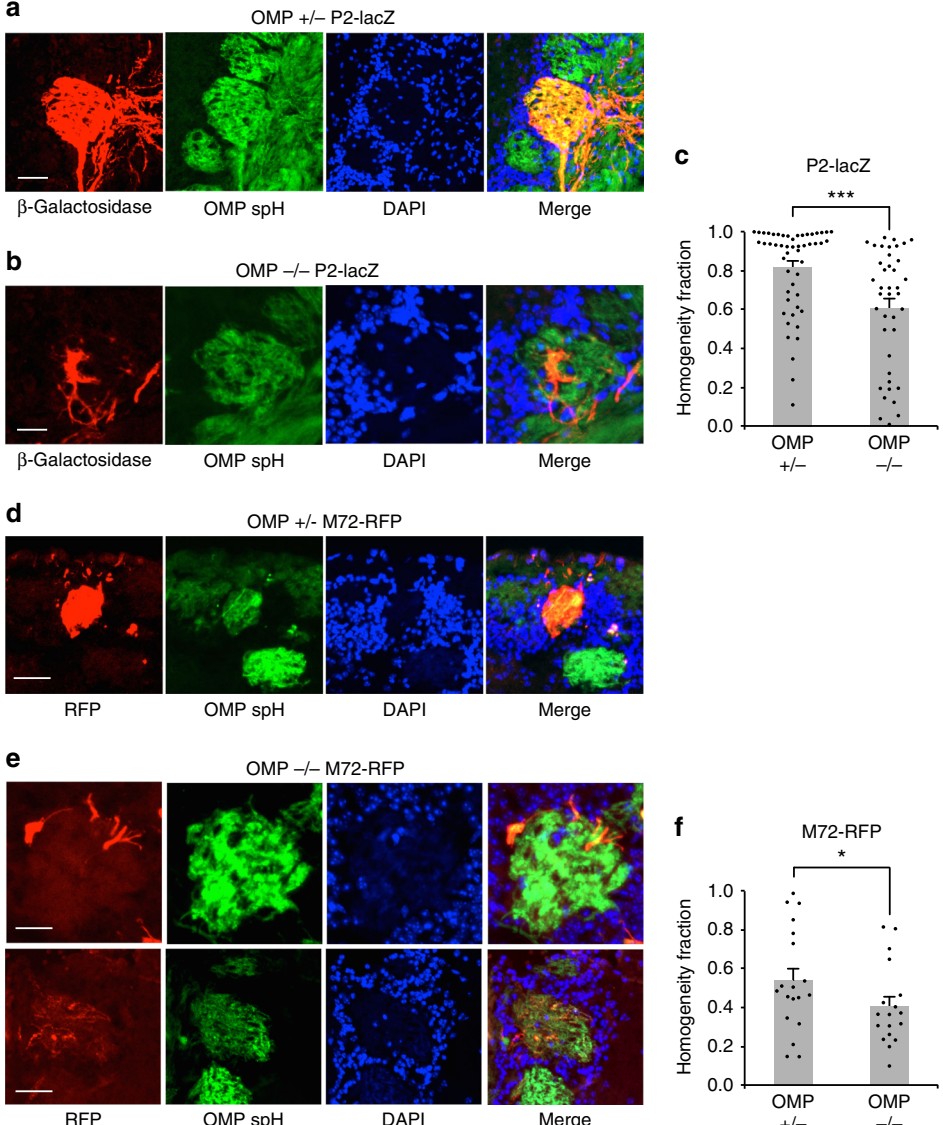

**Fig. 4** Anatomical glomerular heterogeneity in *OMP⁻/⁻* mice. **a** Example homogeneous glomerulus in a *P2-lacZ OMP⁺/⁻* mouse. P2-positive OSN axons labeled with β-Gal (red), SpH-positive OSN axons (green) and glomerular outline by DAPI-stained nuclei of juxtaglomerular cells (blue). Note the high overlap between the red and green pixels in the merged image. Scale bar, 50 μm. **b** Example heterogeneous glomerulus in a *P2-lacZ OMP⁻/⁻* mouse. P2-positive OSN axons labeled with β-Gal (red) do not overlap entirely with SpH-positive OSN axons (green) and only partially fill the glomerulus, outlined by DAPI-stained nuclei of juxtaglomerular cells (blue). Scale bar, 50 μm. **c** Quantification of glomerular homogeneity (P2-lacZ) in the *OMP⁺/⁻* versus *OMP⁻/⁻* mice. Homogeneity fraction: 0.82 ± 0.03 (mean ± SEM, *n* = 52 glomeruli) in *OMP⁺/⁻* mice, and 0.61 ± 0.05 (mean ± SEM, n = 40 glomeruli) in *OMP⁻/⁻* animals. Scale bar, 50 μm. **d**, **e** Same as (**a**) and (**b**) an M72-RFP example glomerulus in the *OMP⁺/⁻* and *OMP⁻/⁻* genetic backgrounds. Scale bar, top 25 μm, bottom 40 μm. **f** Quantification of glomerular homogeneity in *M72-RFP OMP⁺/⁻* versus *OMP⁻/⁻* mice. Homogeneity fraction: 0.54 ± 0.06 (mean ± SEM, *n* = 19 glomeruli) in *OMP⁺/⁻* mice, and 0.41 ± 0.05 (mean ± SEM, n = 18 glomeruli) in *OMP⁻/⁻* animals; *p < 0.05, ***p < 0.001, Kolmogorov–Smirnov test

duplicates of P2 glomeruli in both the *OMP⁻/⁻* and *OMP⁺/⁻* backgrounds in fixed tissue slices. We observed one or two P2 glomeruli per hemifield in the *OMP⁺/⁻* mice (Fig. 5c), and sometimes more than two P2 glomeruli in the *OMP⁻/⁻* mouse (Fig. 5d). In Fig. 5d, multiple glomeruli in different tissue sections contain P2 axons (red), often in microdomains. Overall in the P2 line, glomerular duplicates occurred significantly more frequently in the *OMP⁻/⁻* than in the *OMP⁺/⁻* background, displaying 2.39 ± 0.16 (*n* = 28 hemibulbs) and 1.82 ± 0.14 (*n* = 28 hemibulbs) glomeruli per hemifield, respectively (*p* < 0.05, Wilcoxon ranked-sum test; Fig. 5e).

These results indicate that OMP null leads to heterogeneity and duplicate glomeruli anatomically. Glomerular duplication was always local, and no gross misguidance of OSN axons at far away

targets was observed in the single OR gene-targeted mice. These data suggest that OMP plays a role in local axon refinement at the glomerulus.

## Discussion

In this study, we present evidence of functional and anatomical glomerular heterogeneity in the main olfactory bulb of mice lacking OMP. The macrostructure of glomeruli was not altered, but within single glomeruli, functional subdomains were activated by odors, and we observed numerous local functionally duplicated glomeruli. Moreover, we identified anatomical subdomains in gene-targeted mice in which OSNs expressing a particular OR also co-express a fluorescent protein. *OMP⁻/⁻* mice tended to

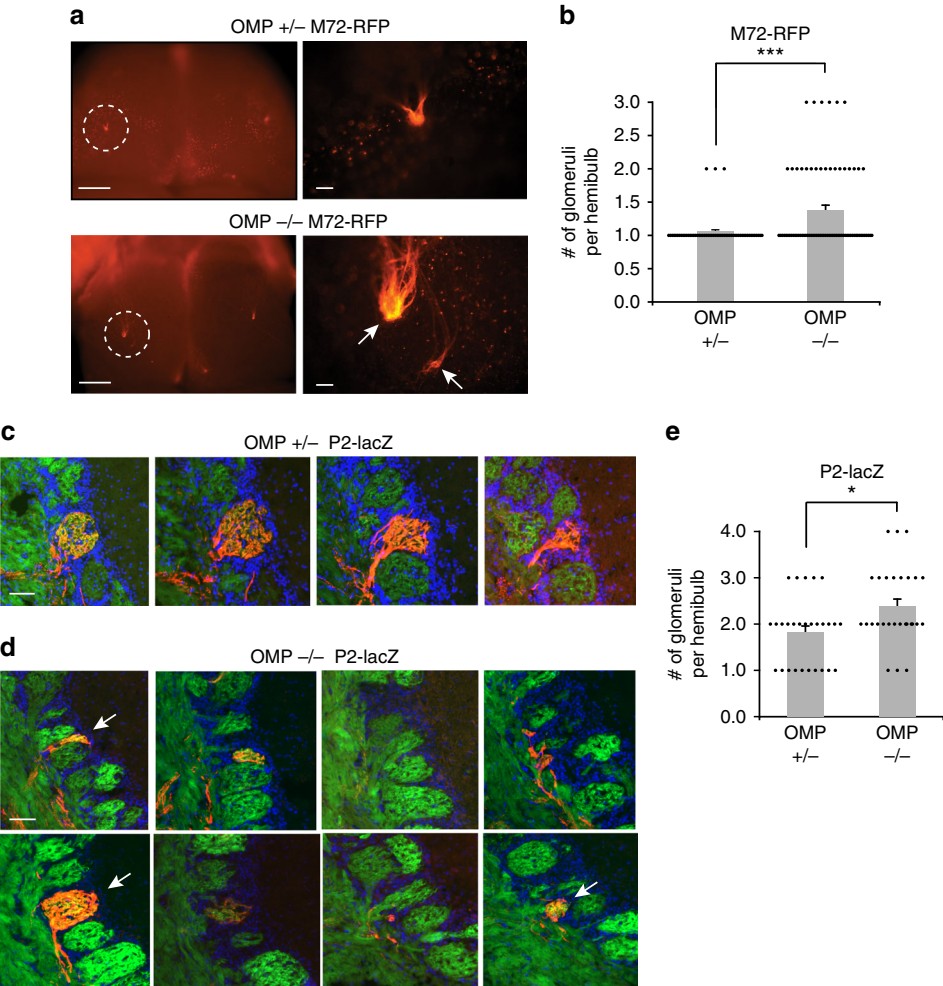

**Fig. 5** Duplicate glomeruli in the *OMP*$^{-/-}$ mice. **a** (Left) Whole mount of the OB dorsal surface in a M72-RFP *OMP*$^{+/-}$ (Top) and *OMP*$^{-/-}$ (Bottom) mouse with a the M72 glomerulus fluorescently labeled. Scale bar, 500 μm. (Right) Zoom-in of circled region. Note the duplicated glomeruli in the *OMP*$^{-/-}$ mouse (arrows). Scale bar, 100 μm. **b** Quantification of duplicate glomeruli in *M72-RFP* mice for *OMP*$^{+/-}$ and *OMP*$^{-/-}$ mice. Shown are 1.05 ± 0.03 glomeruli per hemifield (mean ± SEM, n = 56 hemibulbs) in *OMP*$^{+/-}$ mice and 1.36 ± 0.07 glomeruli per hemifield (mean ± SEM, n = 76 hemibulbs) in *OMP*$^{-/-}$ mice. **c** Serial coronal sections through individual glomeruli in one hemibulb of a *P2-lacZ OMP*$^{+/-}$ mouse. P2-positive OSN axons are labeled with β-Gal (red), OSN axons with SpH (green), and nuclei of juxtaglomerular cells surrounding glomeruli by DAPI (blue). Scale bar, 50 μm. **d** Serial coronal sections of a multiple glomeruli in one hemibulb of a *P2-lacZ OMP*$^{-/-}$ mouse. Arrows indicate distinct glomeruli and fluorescent labels as in (**c**). Scale bar, 50 μm. **e** Quantification of duplicate glomeruli in *P2-lacZ* mice for *OMP*$^{+/-}$ and *OMP*$^{-/-}$ mice. Shown are 1.82 ± 0.14 glomeruli per hemifield (mean ± SEM, n = 28 hemibulbs) in *OMP*$^{+/-}$ mice and 2.39 ± 0.16 glomeruli per hemifield (mean ± SEM, n = 28 hemibulbs) in *OMP*$^{-/-}$ mice. *P < 0.05, ***p < 0.001, Wilcoxon rank-sum test

exhibit anatomically and functionally heterogeneous glomeruli more frequently than *OMP*$^{+/-}$ mice. We also observed duplicate glomeruli in close proximity of one another, but no ectopic glomeruli were found far away from the target site indicating failure of local axon pruning, rather than radical changes in axon guidance. These findings indicate that OMP is necessary for local glomerular refinement, and the lack of OMP leads to an 'immature' glomerular map.

High-resolution imaging in our study has revealed, for the first time, functional heterogeneity within an individual glomerulus. Such heterogeneity is rare in wild-type animals, akin to what we report in *OMP*$^{+/-}$ mice, potentially explaining why a previous study using multiphoton imaging in this mouse did not uncover signs of it[13]. We also note that a substantial number of odorants may be needed to observe glomerular heterogeneity, since differential activation of individual microdomains is necessary for their identification. This suggests that the frequency of heterogeneous glomeruli we report here in the *OMP*$^{-/-}$ mice represents a lower bound.

A recent report noted that tuning curves in *OMP*$^{-/-}$ mice are wider than in *OMP*$^{+/-}$ mice[36]. We propose that the functional and anatomical phenotype presented here accounts for increased tuning widths. Similar to Kass et al.[36], we found that with lower-resolution widefield imaging, a single glomerulus responds to more odors and, strikingly, on average a single odor evokes nearly double the number of responding glomeruli in *OMP*$^{-/-}$ mice. This response pattern is explained by microdomain responses within single glomeruli that are not resolvable with widefield imaging conditions.

Our anatomical data support the explanation that functional microdomains recorded in vivo are due to the convergence of axons that express different ORs into a single glomerulus. We often observed partially labeled glomeruli in gene-targeted mice and infer that the remainder of these glomeruli contain axons expressing different ORs. An alternative possibility, that the same OR has widely different odor responses and segregates differently within a glomerulus, would be a surprising finding requiring a

fundamental rewriting of odor coding rules. Yet another possible explanation is related to differences in concentration of the stimulus as seen by different regions of the glomerulus, perhaps due to differences in odorant sampling at the olfactory epithelium. This possibility is unlikely since different odorants should have similar patterns of differential activation, which is not seen in our data. An additional possibility is that individual OSNs express multiple receptors in the absence of OMP[26], causing differences among OSNs innervating a single glomerulus. This also seems unlikely as the tuning widths of single microdomains in the OMP-null mice, and whole glomerular responses in the $OMP^{+/-}$ mice, were similar, indicating that the one neuron–one receptor rule is intact in $OMP^{-/-}$ mice.

A potential alternative explanation of our results is that the expression of spH, rather than the deletion of OMP, causes mistargeting of the OSN axons. SpH may perturb neurotransmitter release, thus altering the activity of OSNs and resulting in mixed glomeruli. Several observations argue strongly against this possibility. First, we found no relation between heterogeneous glomeruli and the levels of spH. Second, spH expression levels (judged by fluorescence intensity) were not different in $OMP^{+/-}$ and $OMP^{-/-}$ mice, yet there was a substantial difference in the degree of heterogeneity. Third, the fraction of heterogeneous glomeruli we observed in $OMP^{+/-}$ mice (which expressed spH) was very similar to anatomical estimates in $OMP^{+/+}$ mice. For example, Zou et al.[7] found the fraction of heterogeneous M71 glomeruli in adult mice to be around ~8% and that of M72 glomeruli to be ~18%. Fourth, our experiments with OMP-GCaMP3 mice (which are $OMP^{+/+}$) revealed very similar degree of glomerular heterogeneity as $OMP^{+/-}$ mice expressing spH, arguing against the possibility that spH expression in OSNs alters glomerular targeting.

Together, the above observations strongly support the notion that the microdomains observed are due to convergence of multiple types of OSNs, as defined by the specific ORs they express, to individual glomeruli.

OMP is thought to be involved in the OSN signal transduction pathway[30], and other studies in the OMP null have revealed slower response dynamics, decreased sensitivity, and lowered specificity to odors, similar to those of perinatal OSNs[26]. Slow dynamics of response were also reported by others[36], but Kass et al.[28] report that slowed activity does not lead to decreased total activity, indicating that lack of activity may not be the culprit.

Anatomically heterogeneous and duplicate glomeruli have been reported in adolescent mice[7] indicating that glomerular refinement occurs in young mice and homogeneous glomeruli emerge in adulthood. Moreover, activity is important in refinement and maintenance of the glomerular map[7,14]. In both studies, OSNs experienced decreased level of inputs from environment throughout postnatal maturation, and, importantly, lower spontaneous activity compared to wild-type conditions. OMP involvement in signal transduction may lead, in principle, to a very similar phenotype during postnatal development of the odor map, since the onset and offset kinetics of OSNs are slowed down, as is the recovery from odor adaptation. This suppression of activity has multiple downstream effects, including disrupting intraglomerular maps[44].

An interesting case of heterogeneous glomeruli is that in the monoclonal M71 mouse that expresses the M71 OR in ~95% of its OSNs[45]. This mouse displays spherical glomeruli that tile the OB surface, rather than a single massive glomerulus. The authors found that single glomeruli could be anatomically heterogeneous exhibiting axons from both a gene-targeted OSN type and untargeted, non-labeled OSNs. Odor responses in this mouse are observed throughout the entire glomerular field, and discrete glomerular responses are rare. We propose that differential

activity is important for glomerular segregation. In the knockout, OMP role in the signal transduction pathway leads to slow, blurred dynamics[30,36] that may not allow for local differentiation. This is supported by the enrichment paradigm of McGann and colleagues[28] in which OMP-spH-null odor responses become sparser after olfactory enrichment[28]. We posit that enrichment leads to increased activity that allows for more efficient differentiation between OSN types. The resulting reduction in glomerular heterogeneity leads to fewer responding glomeruli per odor.

Another possibility is that OMP is directly involved in axon guidance in OSNs. This hypothesis is prompted by the presence of OMP in OSN axons, where it may regulate the level of cAMP and therefore affect axon path finding. The OB glomerular map is dependent on varying levels of cAMP controlled by a non-olfactory G protein (Gs) in the OSN axons[33,34] and by adenylyl cyclase III (Adcy3)[46,47]. It is thought that the activity profile of each OR can modulate the cAMP levels locally via Gs and in turn cAMP levels can affect expression of axon guidance molecules like neuropilin 1 (Nrp1)[34,48]. The cAMP level-dependent regulation of glomerular positioning appears to be most relevant for global anterior-posterior positioning[49,50].

An additional step of local segregation of axons is thought to occur by different mechanisms that may rely on cell adhesion molecules[51–53]. For example, the level of neuronal activity determines the degree of expression of the homophilic adhesive proteins Kirrel2/Kirrel3[53], which may aid local coalescence of homotypic axons. If activity levels are perturbed, then expression patterns of such proteins could be altered, leading to inadequate segregation. Since the level of cAMP signaling determines the expression of these 'Type II molecules', OMP could be involved in local axon guidance in the OB by regulating cAMP levels. This hypothesis is further corroborated by recent work connecting OMP to cAMP levels. Elevating intracellular cAMP levels by adding phosphodiesterase inhibitors restored normal response kinetics in $OMP^{-/-}$ mice[30]. This implies that OMP acts upstream of cAMP production in the OSN signal transduction pathway, and that loss of OMP decreases cAMP levels, perturbing the normal expression of 'Type II' molecules, thus preventing proper glomerular convergence. OMP may also be involved in the feedback process that stabilizes OR choice and prevents OR gene switching. Lyons et al.[54] have shown that Adcy3 is essential in the OR stabilization process, and thus OMP may play an auxiliary role in the timely stabilization of the expression of the chosen OR. In this scenario, even if the olfactory sensory neurons express a single OR (as the imaging data suggest), they may not express the OR that guided these axons to the specific glomerulus, explaining the existence of functional microdomains.

Our findings demonstrate that OMP is necessary for efficient segregation of axons in their target glomeruli and fulfillment of the one receptor–one glomerulus rule. Functionally, $OMP^{-/-}$ mice presented glomeruli populated by microdomains of differing odor sensitivities, whereas in $OMP^{+/-}$ mice, odor responses within a glomerulus tended to be homogeneous. Using gene-targeted mice, we found that $OMP^{-/-}$ mice exhibited both anatomical microdomains within glomeruli and duplicate glomeruli, and these phenotypes were infrequent in $OMP^{+/-}$ mice. Our results indicate that OMP plays an important function in axonal refinement of the adult OB glomerular map.

## Methods

**In vivo imaging**. Fourteen $OMP^{-/-}$, eight $OMP^{+/-}$ spH mice and five OMP-GCaMP3 mice were anesthetized using a cocktail of ketamine/xylazine (90 mg/kg +9 mg/kg), supplemented every 45 min (30 mg/kg+3 mg/kg), and their heads fixed to a thin metal plate with acrylic glue. The dorsal aspect of the olfactory bulb was exposed through a small cranial window covered with 1.2% low melting point

agarose and a glass coverslip to keep the tissue moist and to minimize motion artifacts. Heartbeat, respiratory rate, and lack of pain reflexes were monitored throughout the experiment. All animal procedures conformed to the National Institutes of Health (NIH) guidelines and were approved by Harvard University's Animal Care and Use Committee.

**Odorant stimulation**. A custom odor delivery machine was built to deliver up to 100 stimuli automatically and in any desired sequence under computer control of solenoid valves (AL4124 24 VDC, Industrial Automation Components). Pure chemicals and mixtures were obtained from Sigma and International Flavors and Fragrances. Odorants were diluted 100-fold into mineral oil and placed in blood collection tubes (Vacutainer, #366431) loaded on a custom-made rack and sealed with a perforated rubber septum circumscribing two blunt end needles (Mcmaster, #75165A754). Fresh air was pumped into each tube via one needle by opening the corresponding solenoid valve. The mixed odor stream exited the tube through the other needle and was delivered at ~0.5 l/min via Teflon-coated tubing to the animal's snout. For each stimulus, 20 s of odorant presentation was preceded and followed by 20 s of fresh air. At least 30 to 60 s of additional fresh air between stimuli was allowed. Each stimulus was delivered 1–3 times. A list of the odorants used in our widefield (99) and two-photon (32) or (15) experiments is provided in Supplementary Table 1. For OMP-GCaMP3 experiments, stimuli were delivered through a custom-built 16 channel olfactometer controlled by custom-written software in LabView. Odorants were maintained at a nominal volumetric concentration of 16% (v/v) in mineral oil and further diluted 16 times with air for a final concentration of 1%. Odors were presented for 5 s with an inter-stimulus interval of at least 40 s. Odor stimulation was repeated at least three times for each field of view.

**Multiphoton imaging**. A custom-built two-photon microscope was used for in vivo imaging. SpH signals were imaged with a water immersion objective (20×, 0.95 NA, Olympus) at 910 nm using a Mira 900 Ti:Sapphire laser (8 W Verdi pump laser) with a 150 fs pulse width and 76 MHz repetition rate. The shortest possible optical path was used to bring the laser onto a galvanometric mirror scanning system (6215HB, Cambridge Technologies). The scanning system projected the incident laser beam through a scan lens and tube lens to backfill the aperture of an Olympus 20×, 0.95 NA objective. A Hamamatsu R3896 photomultiplier was used as photo-detector and a Pockels cell (350-80 BK and 302RM driver, Con Optics) as beam power modulator. The current output of the PMT was transformed to voltage, amplified (SR570, Stanford Instruments) and digitized using a data acquisition board that also controlled the scanning (PCI 6110, National Instruments). Image acquisition and scanning (2-5 Hz) were controlled by custom-written software in Labview.

**Signal detection and analysis**. Odor responses were calculated as average change in fluorescence intensity divided by baseline (pre-odor) intensity using software written in IgorPro or MATLAB. Regions of interest (ROIs) were manually selected based on anatomy and odor responses. Care was taken to avoid selecting ROIs with overlapping neuropil. To facilitate detection of responding glomeruli, we calculated a ratio image for each odor (average of images in odor period minus average of images in pre-stimulus period, normalized by the pre-stimulus average). We further obtained a maximum pixel projection of all odor responses, assigning to each pixel in the field of view the maximum response amplitude across the odorants sampled—this allowed us to visually identify odor-responsive regions. These responsive ROIs mapped to individual glomeruli in the fluorescence image and were selected for further analysis.

We used PCA to identify pixels within the glomerulus that have similar response dynamics. Pixels above the signal threshold (2 SDs from baseline fluctuations) were projected on the plane determined by two of the first three principal components (PCs) revealing functional clusters. We found that ~50% (53 ± 3.9%) of the variance in the data was explained by the first 3 PCs. Although additional PCs were needed to explain the data, we found that the first 3 PCs were sufficient to define glomerular heterogeneity. To determine the spatial location of these clusters, we correlated the average response spectrum of each identified functional cluster to the odor response spectra of each pixel in the field of view. In a subset of experiments, the same analysis was extended to larger field of views (FOVs), across FOVs obtained from the same animal, as well as across animals within similar locations on the bulb surface.

To quantify the fraction of heterogeneous glomeruli we used two criteria: (1) consistent spatial segregation (no overlap) in odor responses to different stimuli in our panel, involving the same microdomains within the glomerulus and (2) separation of odor-evoked responses in distinct functional clusters via PCA which when projected to individual pixels formed spatially distinct (non-overlapping) microdomains in the glomerulus. Average functional separation of clusters within the same anatomical glomerulus (calculated as 1–average correlation) was 0.71 ± 0.04, which was similar to the inter-glomerular separation of 0.63 ± 0.01 (Fig. 2i).

Heterogeneous glomeruli were defined slightly differently for OMP-GCaMP3 experiments since only 15 odors were used (in contrast to 32 for OMP-spH mice). The smaller number of odorants allows for poorer separation of very similar glomeruli or domains (since differences may be missed by chance more easily). A

glomerulus was deemed heterogeneous by observing that only a contiguous subset of its pixels was active in response to particular odors in the panel, while other contiguous pixels within the glomerulus were either not active in response to any of the odors in the panel, or activated by other odors than the pixels forming the first microdomain. This non-active domain could in fact be composed of multiple heterogeneous domains that are not separable with the smaller set of 15 odors.

**Gene-targeted mouse lines**. OMP knockout ($OMP^{-/-}$) mice were generated by replacing both OMP alleles with the with the green synaptic fluorescent reporter SpH[35]. The *P2-IRES-taulacZ* and *M72-IRES- tauRFP[2]* mice were generated using homologous recombination[2].

$OMP$ heterozygous ($OMP^{+/-}$) mice were generated by breeding $OMP^{-/-}$ mice with wild-type C57BL/6 mice (Jackson Laboratory). *P2-IRES-taulacZ* mice were bred with $OMP^{-/-}$ mice to generate *P2-lacZ* mice that were heterozygous or homozygous for the OMP deletion. Similarly, *M72-IRES- tauRFP[2]* were bred with $OMP^{-/-}$ mice to generate *M72-RFP* mice that were heterozygous or homozygous for the OMP gene deletion.

Genotyping for *P2-IRES-taulacZ*, *M72-IRES- tauRFP[2]*, and *OMP SpH* was carried out using DNA purified from toe/tail samples. PCR was conducted using primers in the Jackson Laboratory Mouse Database and EconoTaq PLUS GREEN 2X Master Mix.

All mice were housed under a light/dark cycle of 12/12 h and in accordance with institutional requirements for animal use and care.

**Whole-mount preparation**. $OMP^{-/-}$ or $OMP^{+/-}$ whole-mount brain was prepared by killing the animal, dissecting out the skull, and removing bone to expose the dorsal surface of the OBs.

P2-lacZ $OMP^{-/-}$ or $OMP^{+/-}$ whole-mount brain was prepared by first killing the animal, dissecting out the brain from the skull, fixing the brain in 4% paraformaldehyde (PFA) in 1× phosphate-buffered saline (PBS) for 30–60 min on ice, and rinsing the brain in 1× PBS. The activity of the enzymatic reporter β-galactosidase (β-gal) was revealed by incubating the brain at 37 °C overnight in a 1× PBS solution containing 5 mM potassium-ferricyanide, 5 mM potassium-ferrocyanide, and 1 mg/ml X-gal.

For *M72-RFP* mice, in both $OMP^{-/-}$ and $OMP^{+/-}$ backgrounds, whole-mount brains were prepared by first killing the animal and removing bone to expose the dorsal surface of the OBs. These brains were fixed in 4% PFA in 1× PBS for 30–60 min on ice and then rinsed in 1× PBS.

**Sectioning**. OMP-spH brains were fixed in 4% PFA in 1× PBS overnight and serial coronal sections (50 μm) of the OBs were taken using a Leica VT1000S microtome. Free-floating sections were mounted on VWR Superfrost Plus Micro Slides with VECTASHIELD Mounting Medium, which were then coverslipped and sealed with nail polish.

P2-lacZ and M72-RFP brains were processed for cryo-sectioning. Fresh (unfixed) brains were dissected out of the skull and transferred to a 30% sucrose solution in distilled water at 4 °C until they sank to the bottom of the container. Brains were then embedded in Tissue-Tek optimum cutting temperature (O.C.T.) in a cryo-mold and rapidly frozen on dry ice. Samples were stored at −80 °C and placed in cryostat chamber at −16 °C for 1 h prior to sectioning to achieve temperature equilibration. Serial coronal sections (16 μm) of the OBs were collected onto VWR Superfrost Plus Micro Slides. VECTASHIELD Mounting Medium with DAPI was placed on M72-RFP slides, which were then covered with a glass coverslip and sealed with nail polish. Immunohistochemistry, as described below, was performed on P2-lacZ sections before slides were covered with a glass coverslip with VECTASHIELD Mounting Medium with DAPI and sealed with nail polish.

**Immunohistochemistry**. Following cryo-sectioning, immunohistochemistry was performed on P2-lacZ sections to visualize β-gal signal. Sections were equilibrated in 1× PBS for 5 min at room temperature (RT), fixed in 4% PFA in 1× PBS for 10 min at RT, and washed in 1× PBS for 2 × 3 min at RT. Sections were then incubated in 0.5% Triton-X-100 in 1× PBS for 30 min at RT, washed in 1× PBS for 3 × 5 min at RT, and incubated in 5% normal goat serum in 1× PBS (blocking buffer) for 30 min at RT. Finally, sections were incubated in primary antibodies in blocking buffer overnight at 4 °C, washed in 1× PBS for 3 × 3 min at RT, and incubated in secondary antibodies in blocking buffer for 2 h at RT.

The primary antibody used was rabbit anti-β-gal (Molecular Probes) at a dilution of 1:500 in blocking buffer. The secondary antibody used was goat anti-rabbit conjugated Alexa Fluor 594 (Life Technologies) at a dilution of 1:500 in blocking buffer.

**Imaging**. All whole-mount samples (OMP-spH, P2-lacZ, and M72-RFP) were imaged using the Zeiss Axio Zoom.V16, with varying exposure and magnification settings across samples. Green fluorescent protein (GFP) fluorescence was monitored for OMP-spH whole mounts, reflected brightfield for P2-lacZ whole mounts, and RFP fluorescence for M72-RFP whole mounts.

Serial coronal sections (16 μm) of P2-lacZ and M72-RFP OBs were imaged using the Zeiss Axio Scan.Z1. For P2-lacZ sections, fluorescent channels included

GFP, DAPI, and Alexa Fluor 594, with exposure times of 200 ms, 5 ms, and 50 ms, respectively. For M72-RFP sections, fluorescent channels included GFP, DAPI, and RFP, with exposure times of 200 ms, 5 ms, and 100 ms, respectively. Tiled images of each section were generated at a magnification of ×10 for both P2-lacZ and M72-RFP sections.

**Heterogeneity overlap and glomerular size analysis**. For the quantification of heterogeneity within a single glomerulus, custom software (MATLAB) was used to analyze the overlap between red (β-gal or RFP) signal labeling single odorant receptor identified glomeruli and green (OMP-spH) signal, which labeled all glomeruli. A threshold based on background pixels (selected outside of glomerular regions) was determined, highlighting glomeruli. Glomerular regions of interest were selected in which the number of overlapping red/green pixels, only green pixels, and background pixels was determined. From these pixel measurements, glomerular homogeneity could be expressed as a fraction of the number of overlapping red/green pixels divided by the number of all spH-positive pixels in a single glomerular structure.

## Data availability

The data and code that support the findings of this study are available from the corresponding author upon reasonable request.

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

## Acknowledgements

We thank Vikrant Kapoor, Stephen Santoro, and the Harvard Center for Biological Imaging for advice and assistance for the anatomical experiments. We also thank Allison Nishitani for her comments on the manuscript. P.A. received support from the Harvard College Research Program and Herchel Smith-Harvard Undergraduate Science Research Program. J.D.Z. was supported by an NIH grant (DC015938). This research was supported in part through a grant to V.N.M. from the NIH (DC011291).

## Author contributions

D.F.A.: conceived the project, performed experiments on functional imaging, and wrote the manuscript; A.C.P. and P.A.: performed experiments related to anatomy and wrote the manuscript; E.R.S.: assisted with functional imaging experiments and edited manuscript; J.D.Z.: performed experiments on OMP-GCaMP3 mice; V.N.M.: conceived and supervised the project and wrote the manuscript.

## Additional information

**Competing interests:** A.C.P. is employed by Cohen Veterans Bioscience, a non-profit public charity research organization. The other authors declare no competing interests.

