## [Peer Review File · Nature Communications]

Reviewers' Comments:

Reviewer #1:

Remarks to the Author:

The manuscript by Murphy and colleagues describes an interesting function of olfactory marker protein (OMP) in the axonal targeting of olfactory sensory neurons (OSNs). Specifically the authors show that glomeruli of OMP^{-/-} mice display "microdomains" of differential odor responses that occur in higher frequency than in OMP^{+/-} mice. These functional microdomains likely reflect anatomical microdomains, as depicted by analyses of glomeruli from ORiresReporter mice. The effects are not very strong but appear statistically significant. An obvious question is why the authors did not use OMP^{+/+} mice as controls for their experiments. Given that OMP may act as a phosphodiesterase, affecting cAMP signaling, it is reasonable to assume that OMP protein levels make a difference and that wild type mice may have even fewer microdomains than heterozygote knock out mice. I believe the authors can easily obtain data from OMP^{+/+} mice both for their imaging and for their anatomical studies, which will likely increase the confidence on the observed phenotype. Also, I believe that the authors should entertain the possibility that OMP is involved in the feedback process that stabilizes OR choice and prevents OR gene switching. Lyons et al 2013 has shown that *Adcy3* is essential in the stabilization process, thus OMP may play an auxiliary role in the timely stabilization of the expression of the chosen OR. In that scenario, even if the neurons express a single OR (as the imaging data suggest), they do not express the OR that guided these axons to the specific glomerulus, explaining the existence of functional microdomains. Lineage tracing experiments comparing the stability of OR choice in wild type and OMP KO neurons would answer this, but realistically this experiment would take a year. Thus, I believe that the authors should only discuss this possibility and address this issue in future studies.

Overall, I find this observation interesting and important for the field, for scientific and technical reasons. Scientifically, because the exact role of OMP was never fully elucidated, and this study provides important insight in the role of this protein in the circuitry of the olfactory neurons. Technically, because there is extensive use of the OMP-GFP, OMP-tTA, or OMP-Cre lines, which are essentially OMP KO alleles. If indeed the authors find that the OMP^{+/-} mice have glomerular perturbations in comparison to the OMP^{+/+} controls, the field should switch in using OMPiresGFP, OMPiresCre and OMPiresTAA lines that are available and do not disrupt the OMP gene.

Reviewer #2:

Remarks to the Author:

This manuscript from Albeanu, Murthy and colleagues implicates olfactory marker protein (OMP) as playing a role in convergence of homotypic OSNs into glomeruli, largely using functional imaging of transmitter release from OSN terminals. They show that, in OMP null synaptophluorin-expressing mice, a relatively small fraction of glomeruli are heterogeneous with respect to their OSN inputs, consistent with improper targeting of a fraction of OSNs to their cognate glomeruli. They strengthen this case using P2 and M72 receptor-tagged OSN populations. This is an important finding, and the data presented are compelling and in fact quite beautifully presented. While there is more that could be understood about this phenomenon, overall this is a significant contribution for the field. I have only relatively minor suggestions for improvements to the manuscript. These are detailed below more or less in order of their appearance in the manuscript.

1. Results, p. 5. The first reference to Supplementary Table 1 states that the 98 odorants used in the wide field imaging screen are listed here. They are not (the table only includes the 32 odorants from the 2-P imaging). This should be corrected by adding them to the table.

2. There are some missing details and apparent flaws regarding the PCA analysis used to define glomerular heterogeneity in Figure 2 and related text (see this and the following two comments). First, it is surprising to me that all of the heterogeneity can be cleanly accounted for by precisely three principal components, as implied by the text. For how many glomeruli was this analysis performed? Why would one not observe more than three distinct functional clusters per glomerulus? This should be addressed somehow, and the results of this analysis summarized across multiple glomeruli.
3. Second, for the comparison of functional domains at different imaging plane: this is a nice analysis to do, but technically one cannot call the three functional domains 'the same', since (as I understand it) the PCA is performed de novo at each imaging plane, and so the components cannot be assigned as identical to one another across planes. This would require a different analysis, or some other criteria (such as odorant response spectrum) to compare the domains. This latter analysis would be strengthen the case.
4. Third, re the summary analysis of the fraction of functionally heterogeneous glomeruli (p. 8), I could not find any quantitative criterion by which functional heterogeneity (or not) was determined. It simply says, "using PCA". Using what measure from the PCA? This should be clarified.
5. On p. 10, the sentence reads, "In the M72-RFP line....M72-RFP OMP+/- mice often exhibited filled, homogeneous glomeruli". I believe this is a misstatement and should read "heterogeneous glomeruli", unless I misunderstand the intent of this sentence.
6. A potential alternative explanation for these results is that expression of synaptophluorin, rather than deletion of OMP, causes mistargeting of OSN axons. spH may perturb transmitter release in some subtle way which could impact convergence. While this might seem unlikely, this explanation is no more speculative than the explanations regarding how OMP might impact targeting. Thus, the possibility should be mentioned in the Discussion.

Reviewer #3:

Remarks to the Author:

This study seeks to address a fundamental question about the targeting of olfactory sensory neurons (OSNs) to glomeruli in the olfactory bulb. How does the system achieve the single odorant receptor (OR)-single glomerulus arrangement wherein OSNs that express single OR send convergent axons onto individual glomeruli? The authors here provide functional and anatomical evidence for a role of the OSN-specific olfactory marker protein (OMP). In mice in which synaptophluorin (spH) replaces the coding sequence for OMP, evidence is provided that removing OMP causes individual glomeruli to be more broadly tuned to odors, consistent with decreased specificity in OSN targeting. In addition, using two-photon laser scanning microscopy combined with functional and anatomical measurements, the authors show that individual glomeruli in OMP $-/-$ mice have significant disruptions in their microarchitecture, wherein OSNs of differing OR-specificities occupy subportions of glomeruli. Moreover, these disruptions appear to be quite local, limited to an area of a few glomeruli. Based on these results, the authors conclude that OMP has an important role in the local refinement of the glomerular olfactory map.

Overall, this is an interesting study that addresses an important and long-standing question about olfactory anatomy that remains largely unresolved. The results presented are also generally quite nice. For example the experiments that examine the odor-tuning profiles of components of the microarchitecture of glomeruli make a compelling and direct case for mistargeting of OSNs in OMP $-/-$ mice. The dual use of functional and anatomical methods is also a strength, and most of their results

are basically convincing. My main concern is that, while the study shows extremely well that there is mistargeting of OSNs in their OMP $-/-$ mice, the results do not shed significant light on what OMP is actually doing. There is a good possibility that its role is quite indirect, through changes in activity levels in OSNs. In addition, the authors need to better exclude the possibility that the OMP $-/-$ effects are not due to the replacement protein spH, and OMP's role in strictly "local" refinement of OSN targeting also needs to be better established.

Major concerns

1. My most important concern pertains to the extent to which this paper adds to the existing body of knowledge about the mechanisms underlying the single OR-single glomerulus rule for OSN targeting. It is already known, for example, that activity of OSNs can impact targeting of OSNs, and, as the authors point out, their results here could be explained by disruptions in OSN activity by OMP $-/-$. If this is case, the role of OMP would be indirect and of the type that could be explained by a quite large number of different proteins and ion channels that impact neural activity. OMP could have a more direct role, but the present study does not provide supportive evidence for this.
2. Making the case that OMP has a direct role in targeting of OSNs would also require that the authors better address the possibility that the effects of OMP $-/-$ might have been due to expression of the replacement protein spH. If spH has any impact on glutamate release from OSNs, this could in principle impact OSN targeting, which could require coincident activation of OSNs and post-synaptic cells in glomeruli. It is not clear to me that the possibility of synaptotagmin expression impacting glutamate release from OSNs (or another critical such function in OSNs) has been satisfactorily addressed in the literature.
3. That mistargeting of OSNs in the OMP $-/-$ is restricted to a local neighborhood of a few glomerular diameters is not completely convincing. Evidence is provided in Figure 3 that functionally fragmented glomeruli were often observed next to functionally matched glomeruli, which is consistent with aberrant local refinement. However, the basis for the statement in the Discussion that "no ectopic glomeruli were found far away from the target site" – which would be consistent with ONLY local changes – is not clear.

Reviewer #1 (Remarks to the Author):

The manuscript by Murphy and colleagues describes an interesting function of olfactory marker protein (OMP) in the axonal targeting of olfactory sensory neurons (OSNs). Specifically the authors show that glomeruli of OMP^{-/-} mice display “microdomains” of differential odor responses that occur in higher frequency than in OMP^{+/-} mice. These functional microdomains likely reflect anatomical microdomains, as depicted by analyses of glomeruli from ORiresReporter mice. The effects are not very strong but appear statistically significant. An obvious question is why the authors did not use OMP^{+/+} mice as controls for their experiments. Given that OMP may act as a phosphodiesterase, affecting cAMP signaling, it is reasonable to assume that OMP protein levels make a difference and that wild type mice may have even fewer microdomains than heterozygote knock out mice. I believe the authors can easily obtain data from OMP^{+/+} mice both for their imaging and for their anatomical studies, which will likely increase the confidence on the observed phenotype. Also, I believe that the authors should entertain the possibility that OMP is involved in the feedback process that stabilizes OR choice and prevents OR gene switching. Lyons et al 2013 has shown that *Adcy3* is essential in the stabilization process, thus OMP may play an auxiliary role in the timely stabilization of the expression of the chosen OR. In that scenario, even if the neurons express a single OR (as the imaging data suggest), they do not express the OR that guided these axons to the specific glomerulus, explaining the existence of functional microdomains. Lineage tracing experiments comparing the stability of OR choice in wild type and OMP KO neurons would answer this, but realistically this experiment would take a year. Thus, I believe that the authors should only discuss this possibility and address this issue in future studies.

Overall, I find this observation interesting and important for the field, for scientific and technical reasons. Scientifically, because the exact role of OMP was never fully elucidated, and this study provides important insight in the role of this protein in the circuitry of the olfactory neurons. Technically, because there is extensive use of the OMP-GFP, OMP-tTA, or OMP-Cre lines, which are essentially OMP KO alleles. If indeed the authors find that the OMP^{+/-} mice have glomerular perturbations in comparison to the OMP^{+/+} controls, the field should switch in using OMPiresGFP, OMPiresCre and OMPiresTA lines that are available and do not disrupt the OMP gene.

We thank the Reviewer for these helpful comments and the support for the work. We address the questions/concerns raised.

First, the Reviewer mentions that the effects are not very strong. We respectfully disagree, since heterogeneity is found in ~25% of glomeruli (in other words, 1 in 4!). Since mice have more than 2000 glomeruli in each olfactory bulb, more than 500 of these will be heterogeneous – surely a strong effect! In addition, we suspect that the fraction of heterogeneous glomeruli is actually higher than we report here. This is because in order to reveal the presence of the heterogeneous glomeruli, we need both to: 1) trigger odor evoked responses from these glomeruli; 2) do this in a differential manner across the two or more micro-domains within a

Reviewers' Comments:

Reviewer #1:

Remarks to the Author:

Murphy and colleagues responded to the issues raised by the reviewers in a satisfactory fashion. One issue I had regarding the use of OMP^{+/+} as control mice for their experiments was not addressed. I understand that the OMP gene is driving their reporter, thus calcium imaging experiments cannot be performed in a ^{+/+} background, however the authors provided also anatomical data showing the existence of microdomains. Thus, they could have easily explored ORiresGFP; OMP^{+/+} or ORiresGFP; OMP^{+/-} for increased microdomains in the heterozygote OMP mutants. However, I agree that given the lower frequency of microdomains in the OMP^{-/+} controls, this experiment is not necessary. Thus, I support publication of this very interesting article.

Reviewer #2:

Remarks to the Author:

The authors have addressed the issues raised in my initial review, and while I agree with the limitations in interpretation raised by the other reviewers, I think this is a very nice contribution that will be quite useful to the field.

Reviewer #3:

Remarks to the Author:

The additional new analyses and discussion in the paper adequately address most of my prior concerns.